# DISCOVERING PARAMETRIC ACTIVATION FUNCTIONS

## ABSTRACT

Recent studies have shown that the choice of activation function can significantly affect the performance of deep learning networks. However, the benefits of novel activation functions have been inconsistent and task dependent, and therefore the rectified linear unit (ReLU) is still the most commonly used. This paper proposes a technique for customizing activation functions automatically, resulting in reliable improvements in performance. Evolutionary search is used to discover the general form of the function, and gradient descent to optimize its parameters for different parts of the network and over the learning process. Experiments with four different neural network architectures on the CIFAR-10 and CIFAR-100 image classification datasets show that this approach is effective. It discovers both general activation functions and specialized functions for different architectures, consistently improving accuracy over ReLU and other recently proposed activation functions by significant margins. The approach can therefore be used as an automated optimization step in applying deep learning to new tasks.

## 1 INTRODUCTION

The rectified linear unit ($\text{ReLU}(x) = \max\{x, 0\}$) is the most commonly used activation function in modern deep learning architectures (Nair & Hinton, 2010). When introduced, it offered substantial improvements over the previously popular tanh and sigmoid activation functions. Because ReLU is unbounded as $x \to \infty$, it is less susceptible to vanishing gradients than tanh and sigmoid are. It is also simple to calculate, which leads to faster training times.

Activation function design continues to be an active area of research, and a number of novel activation functions have been introduced since ReLU, each with different properties (Nwankpa et al., 2018). In certain settings, these novel activation functions lead to substantial improvements in accuracy over ReLU, but the gains are often inconsistent across tasks. Because of this inconsistency, ReLU is still the most commonly used: it is reliable, even though it may be suboptimal.

The improvements and inconsistencies are due to a gradually evolving understanding of what makes an activation function effective. For example, Leaky ReLU (Maas et al., 2013) allows a small amount of gradient information to flow when the input is negative. It was introduced to prevent ReLU from creating dead neurons, i.e. those that are stuck at always outputting zero. On the other hand, the ELU activation function (Clevert et al., 2015) contains a negative saturation regime to control the forward propagated variance. These two very different activation functions have seemingly contradicting properties, yet each has proven more effective than ReLU in various tasks.

There are also often complex interactions between an activation function and other neural network design choices, adding to the difficulty of selecting an appropriate activation function for a given task. For example, Ramachandran et al. (2018) warned that the scale parameter in batch normalization (Ioffe & Szegedy, 2015) should be set when training with the Swish activation function; Hendrycks & Gimpel (2016) suggested using an optimizer with momentum when using GELU; Klambauer et al. (2017) introduced a modification of dropout (Hinton et al., 2012) called alpha dropout to be used with SELU. These results suggest that significant gains are possible by designing the activation function properly for a network and task, but that it is difficult to do so manually.

This paper presents an approach to automatic activation function design. The approach is inspired by genetic programming (Koza, 1992), which describes techniques for evolving computer programs to solve a particular task. In contrast with previous studies (Bingham et al., 2020; Ramachandran et al., 2018; Liu et al., 2020; Basirat & Roth, 2018), this paper focuses on automatically discovering activation functions that are parametric. Evolution discovers the general form of the function, while gradient descent optimizes the parameters of the function during training. The approach,

Table 1: The operator search space consists of basic unary and binary functions as well as existing activation functions (Appendix D). $\sigma(x) = (1 + e^{-x})^{-1}$. The unary operators bessel_i0e and bessel_i1e are the exponentially scaled modified Bessel functions of order 0 and 1, respectively.

| | | | Unary | | | | Binary | |
|---|---|---|---|---|---|---|---|---|
| 0 | $|x|$ | $\text{erf}(x)$ | $\tanh(x)$ | $\text{arcsinh}(x)$ | $\text{ReLU}(x)$ | $\text{Softplus}(x)$ | $x_1 + x_2$ | $x_1^{x_2}$ |
| 1 | $x^{-1}$ | $\text{erfc}(x)$ | $e^x - 1$ | $\text{arctanh}(x)$ | $\text{ELU}(x)$ | $\text{Softsign}(x)$ | $x_1 - x_2$ | $\max\{x_1, x_2\}$ |
| $x$ | $x^2$ | $\sinh(x)$ | $\sigma(x)$ | $\text{bessel\_i0e}(x)$ | $\text{SELU}(x)$ | $\text{HardSigmoid}(x)$ | $x_1 \cdot x_2$ | $\min\{x_1, x_2\}$ |
| $-x$ | $e^x$ | $\cosh(x)$ | $\log(\sigma(x))$ | $\text{bessel\_i1e}(x)$ | $\text{Swish}(x)$ | | $x_1/x_2$ | |

called PANGAEA (Parametric ActivatioN functions Generated Automatically by an Evolutionary Algorithm), discovers general activation functions that improve performance overall over previously proposed functions. It also produces specialized functions for different architectures, such as Wide ResNet, ResNet, and Preactivation ResNet, that perform even better than the general functions, demonstrating its ability to customize activation functions to architectures.

## 2 RELATED WORK

Prior work in automatic activation function discovery includes that of Ramachandran et al. (2018), who used reinforcement learning to design novel activation functions. They discovered multiple functions, but analyzed just one in depth: $\text{Swish}(x) = x \cdot \sigma(x)$. Of the top eight functions discovered, only Swish and $\max\{x, \sigma(x)\}$ consistently outperformed ReLU across multiple tasks, suggesting that improvements are possible but often task specific.

Bingham et al. (2020) used evolution to discover novel activation functions. Whereas their functions had a fixed graph structure, PANGAEA utilizes a flexible search space that implements activation functions as arbitrary computation graphs. PANGAEA also includes more powerful mutation operations, and a function parameterization approach that makes it possible to further refine functions through gradient descent.

Liu et al. (2020) evolved normalization-activation layers. They searched for a computation graph that replaced both batch normalization and ReLU in multiple neural networks. They argued that the inherent nonlinearity of the discovered layers precluded the need for any explicit activation function. However, experiments in this paper show that carefully designed parametric activation functions can in fact be a powerful augmentation to existing deep learning models.

Finally, Basirat & Roth (2018) used a genetic algorithm to discover task-specific piecewise activation functions. They showed that different functions are optimal for different tasks. However, the discovered activation functions did not outperform ELiSH and HardELiSH, two hand-designed activation functions proposed in the same paper (Basirat & Roth, 2018). The larger search space in PANGAEA affords evolution extra flexibility in designing activation functions, while the trainable parameters give customizability to the network itself, leading to consistent, significant improvement.

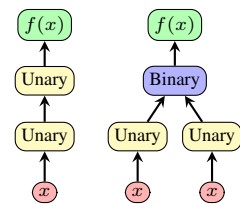

Figure 1: Random activation function initialization. The initial population consists of random samples of two kinds of computation graphs, randomly initialized with the operators in Table 1. In this manner, the search starts with simple graphs and gradually expands to more complex forms.

## 3 THE PANGAEA METHOD

### 3.1 REPRESENTING AND MODIFYING ACTIVATION FUNCTIONS

Activation functions are represented as computation graphs in which each node is a unary or a binary operator (Table 1). The activation functions are implemented in TensorFlow (Abadi et al., 2016), and safe operator implementations are chosen when possible (e.g. the binary operator $x_1/x_2$ is implemented as `tf.math.divide_no_nan`, which returns 0 if $x_2 = 0$). The operators in Table 1 were chosen to create a large and expressive search space that contains activation functions unlikely to be discovered by hand. Operators that are periodic (e.g. $\sin(x)$) and operators that contain repeated asymptotes were not included; in

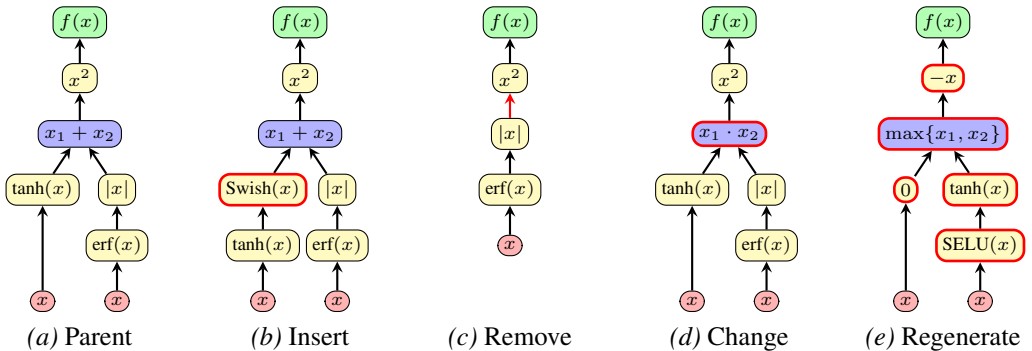

*(a)* Parent     *(b)* Insert     *(c)* Remove     *(d)* Change     *(e)* Regenerate

Figure 2: Evolutionary operations on activation functions. In an 'Insert' mutation, a new operator is inserted in one of the edges of the computation graph, like the $\text{Swish}(x)$ in *(b)*. In a 'Remove' mutation, a node in the computation graph is deleted, like the addition in *(c)*. In a 'Change' mutation, an operator at a node is replaced with another, like addition with multiplication in *(d)*. These first three mutations are useful in refining the function locally. In contrast, in a 'Regenerate' mutation *(e)*, every operator in the graph is replaced by a random operator, thus increasing exploration.

preliminary experiments they often caused training instability. All of the operators have domain $\mathbb{R}$, making it possible to compose them arbitrarily.

PANGAEA begins with an initial population of $P$ random activation functions. Each function is either of the form $f(x) = \texttt{unary1}(\texttt{unary2}(x))$ or $f(x) = \texttt{binary}(\texttt{unary1}(x), \texttt{unary2}(x))$, as shown in Figure 1. Both forms are equally likely, and the unary and binary operators are also selected uniformly at random. Previous work has suggested that it is difficult to discover high-performing activation functions that have complicated computation graphs (Bingham et al., 2020). The computation graphs in Figure 1 thus represent the simplest non-trivial computation graphs with and without a binary operator.

During the search, all ReLU activation functions in a given neural network are replaced with a candidate activation function. No other changes to the network or training setup are made. The network is trained on the dataset, and the activation function is assigned a fitness score equal to the network's accuracy on the validation set.

Given a parent activation function, a child activation function is created by applying one of four possible mutations (Figure 2). Other possible evolutionary operators like crossover are not used in this paper. All mutations are equally likely with two special cases. If a remove mutation is selected for an activation function with just one node, a change mutation is applied instead. Additionally, if an activation function with greater than seven nodes is selected for mutation, the mutation is a remove mutation, in order to reduce bloat.

**Insert** In an insert mutation, one operator in the search space is selected uniformly at random. This operator is placed on a random edge of a parent activation function graph. In Figure 2b, the unary operator $\text{Swish}(x)$ is inserted at the edge connecting the output of $\tanh(x)$ to the input of $x_1 + x_2$. After mutating, the parent activation function $(\tanh(x) + |\text{erf}(x)|)^2$ produces the child activation function $(\text{Swish}(\tanh(x)) + |\text{erf}(x)|)^2$. If a binary operator is randomly chosen for the insertion, the incoming input value is assigned to the variable $x_1$. If the operator is addition or subtraction, the input to $x_2$ is set to 0. If the operator is multiplication, division, or exponentiation, the input to $x_2$ is set to 1. Finally, if the operator is the maximum or minimum operator, the input to $x_2$ is a copy of the input to $x_1$. When a binary operator is inserted into a computation graph, the activation function computed remains unchanged. However, the structure of the computation graph is modified and can be further altered by future mutations.

**Remove** In a remove mutation, one node is selected uniformly at random and deleted. The node's input is rewired to its output. If the removed node is binary, one of the two inputs is chosen at random and is deleted. The other input is kept. In Figure 2c, the addition operator is removed from the parent activation function. The two inputs to addition, $\tanh(x)$ and $|\text{erf}(x)|$, cannot both be kept. By chance, $\tanh(x)$ is discarded, resulting in the child activation function $|\text{erf}(x)|^2$.

**Change** To perform a change mutation, one node in the computation graph is selected at random and replaced with another operator from the search space, also uniformly at random. Unary operators are always replaced with unary operators, and binary operators with binary operators. Figure 2*d* shows how changing addition to multiplication produces the activation function $(\tanh(x) \cdot |\mathrm{erf}(x)|)^2$.

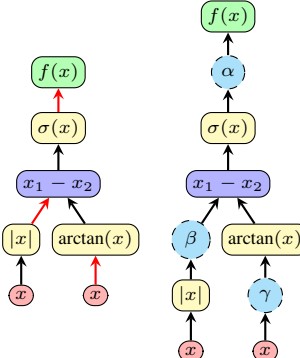

**Regenerate** In a regenerate mutation, every operator in the computation graph is replaced with another operator from the search space. As with change mutations, unary operators are replaced with unary operators, and binary operators with binary operators. Although every node in the graph is changed, the overall structure of the computation graph remains the same. Regenerate mutations are useful for increasing exploration, and are similar in principle to burst mutation and delta coding (Gomez & Miikkulainen, 2003; Whitley et al., 1991). Figure 2*e* shows the child activation function $-\max\{0, \tanh(\mathrm{SELU}(x))\}$, which is quite different from the parent function in Figure 2*a*.

Figure 3: Parameterization of activation functions. In this example, parameters are added to $k = 3$ random edges, yielding the parametric activation function $\alpha\sigma(\beta|x| - \arctan(\gamma x))$.

**Parameterization of Activation Functions** After mutation (or random initialization), activation functions are parameterized (Figure 3). A value $k \in \{0, 1, 2, 3\}$ is chosen uniformly at random, and $k$ edges of the activation function graph are randomly selected. Multiplicative per-channel parameters are inserted at these edges and initialized to one. Whereas evolution is well suited for discovering the general form of the activation function in a discrete, structured search space, parameterization makes it possible to fine-tune the function using gradient descent. The function parameters are updated at every epoch during backpropagation, resulting in different activation functions in different stages of training. As the parameters are per-channel, the process creates different activation functions at different locations in the neural network. Thus, parameterization gives neural networks additional flexibility to customize activation functions.

## 3.2 Discovering Activation Functions with Evolution

Activation functions are discovered by regularized evolution (Real et al., 2019). Initially, $P$ random activation functions are created, parameterized, and assigned fitness scores. To generate a new activation function, $S$ functions are sampled with replacement from the current population. The function with the highest validation accuracy serves as the parent, and is mutated to create a child activation function. This function is parameterized and assigned a fitness score. The new activation function is then added to the population, and the oldest function in the population is removed, ensuring the population is always of size $P$. This process continues until $C$ functions have been evaluated in total, and the top functions over the history of the search are returned as a result.

Any activation function that achieves a fitness score less than a threshold $V$ is discarded. These functions are not added to the population, but they do count towards the total number of $C$ activation functions evaluated for each architecture. This quality control mechanism allows evolution to focus only on the most promising candidates.

To save computational resources during evolution, each activation function is evaluated by training a neural network for 100 epochs using a compressed learning rate schedule (Appendix B). After evolution is complete, the top 10 activation functions from the entire search are reranked. Each function receives an adjusted fitness score equal to the average validation accuracy from two independent 200-epoch training runs using the original learning rate schedule. The top three activation functions after reranking proceed to the final testing experiments.

During evolution, it is possible that some activation functions achieve unusually high validation accuracy by chance. The 100-epoch compressed learning rate schedule may also have a minor effect on which activation functions are optimal compared to a full 200-epoch schedule. Reranking thus serves two purposes. Full training reduces bias from the compressed schedule, and averaging two such runs lessens the impact of activation functions that achieved high accuracy by chance.

## 4 DATASETS AND ARCHITECTURES

The experiments in this paper focus primarily on the CIFAR-100 image classification dataset (Krizhevsky et al., 2009). This dataset is a more difficult version of the popular CIFAR-10 dataset, with 100 object categories instead of 10. Fifty images from each class were randomly selected from the training set to create a balanced validation set, resulting in a training/validation/test split of 45K/5K/10K images.

To demonstrate that PANGAEA can discover effective activation functions in various settings, it is evaluated with three different neural networks. The models were implemented in TensorFlow (Abadi et al., 2016), mirroring the original authors' training setup as closely as possible (Appendix B).

**Wide Residual Network** (WRN-10-4; Zagoruyko & Komodakis, 2016) has a depth of 10 and widening factor of four. Wide residual networks provide an interesting comparison because they are shallower and wider than many other popular architectures, while still achieving good results. WRN-10-4 was chosen because its CIFAR-100 accuracy is competitive, yet it trains relatively quickly.

**Residual Network** (ResNet-v1-56; He et al., 2016a), with a depth of 56, provides an important contrast to WRN-10-4. It is significantly deeper and has a slightly different training setup, which may have an effect on the performance of different activation functions.

**Preactivation Residual Network** (ResNet-v2-56; He et al., 2016b) has identical depth to ResNet-v1-56, but is a fundamentally different architecture. Activation functions are not part of the skip connections, as is the case in ResNet-v1-56. Since information does not have to pass through an activation function, this structure makes it easier to train very deep architectures. PANGAEA should exploit this structure and discover different activation functions for ResNet-v2-56 and ResNet-v1-56.

## 5 RESULTS

**Overview** Separate evolution experiments were run to discover novel activation functions for each of the three architectures. Evolutionary parameters $P = 64$, $S = 16$, $C = 1,000$, and $V = 20\%$ were used since they were found to work well in preliminary experiments.

Figure 4 visualizes progress in these experiments. For all three architectures, PANGAEA quickly discovered activation functions that outperform ReLU. It continued to make further progress, gradually discovering better activation functions, and did not plateau during the time allotted for the experiment. Each run took approximately 2,000 GPU hours on GeForce GTX 1080 GPUs (Appendix C).

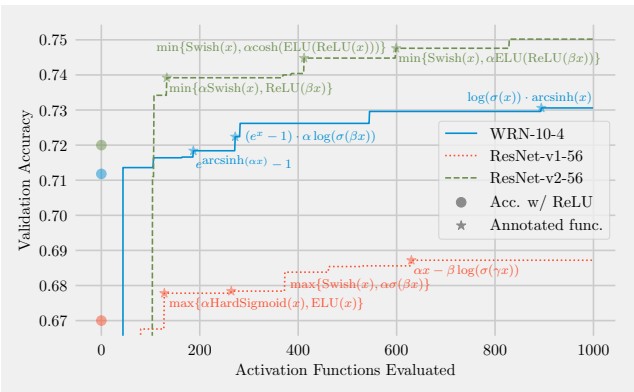

Figure 4: Progress of PANGAEA on three different neural networks. Evolution quickly discovered activation functions that outperform ReLU (shown at $x = 0$), and continued to improve throughout the experiment. The plots show the highest validation accuracy of all activation functions evaluated so far after 100 epochs of training. Notable discovered activation functions are identified with a star and annotated. The improvements over ReLU are meaningful, but the values themselves are not directly comparable to the results in Table 2, which lists test set accuracy after 200 epochs.

Table 2 shows the final test accuracy for the top specialized activation functions discovered by PANGAEA in each run. For comparison, the accuracy of the top general functions discovered in this process are also shown, as well as the accuracy of 28 baseline activation functions. In sum, PANGAEA discovered the best activation function for ResNet-v2-56, the top two activation functions for ResNet-v1-56, and the top three activation functions for WRN-10-4.

**Specialized Activation Functions** For all three architectures, there is at least one baseline activation function that outperforms ReLU by a statistically significant margin. This result already demonstrates the importance of activation function design, and suggests that the common practice of

Table 2: CIFAR-100 test set accuracy shown as a median of ten runs, with mean $\pm$ sample standard deviation in parenthesis. The top accuracy for each architecture is in bold. Asterisks indicate a statistically significant improvement in mean accuracy over ReLU, with * if $p \leq 0.05$, ** if $p \leq 0.01$, and *** if $p \leq 0.001$; $p$-values are from one-tailed Welch's $t$-tests. The ++ or +++ indicate a statistically significant improvement in mean accuracy over all 28 baseline activation functions (Appendix D), with $p \leq 0.01$ or $p \leq 0.001$ in every case, respectively.

| | WRN-10-4 | ResNet-v1-56 | ResNet-v2-56 |
|---|---|---|---|
| **Specialized for WRN-10-4** | | | |
| $\log(\sigma(\alpha x)) \cdot \text{arcsinh}(x)$ | **73.23** (73.16 ± 0.41) *** +++ | 11.15 (19.34 ± 20.14) | 72.05 (64.30 ± 21.32) |
| $\log(\sigma(\alpha x)) \cdot \beta \text{arcsinh}(x)$ | 73.22 (73.20 ± 0.37) *** +++ | 05.78 (18.63 ± 21.04) | 55.40 (45.88 ± 30.70) |
| $-\text{Swish}(\text{Swish}(\alpha x))$ | 72.38 (72.49 ± 0.55) *** | 59.61 (58.86 ± 2.88) | 74.70 (74.71 ± 0.20) * |
| **Specialized for ResNet-v1-56** | | | |
| $\alpha x - \beta \log(\sigma(\gamma x))$ | 70.35 (70.28 ± 0.37) | **70.82** (71.01 ± 0.64) *** ++ | 74.41 (74.35 ± 0.45) |
| $\alpha x - \log(\sigma(\beta x))$ | 70.62 (70.47 ± 0.53) | 70.30 (70.30 ± 0.58) * | 74.73 (74.70 ± 0.23) * |
| $\max\{\text{Swish}(x), 0\}$ | 71.96 (72.10 ± 0.33) ** | 69.46 (69.43 ± 0.69) | 74.97 (74.97 ± 0.25) ** |
| **Specialized for ResNet-v2-56** | | | |
| $\text{Softplus}(\text{ELU}(x))$ | 71.51 (71.36 ± 0.34) | 69.94 (69.96 ± 0.39) | **75.60** (75.61 ± 0.42) *** |
| $\min\{\log(\sigma(x)), \alpha \log(\sigma(\beta x))\}$ | 72.05 (72.04 ± 0.34) ** | 69.63 (69.56 ± 0.48) | 75.20 (75.19 ± 0.39) *** |
| $\text{SELU}(\text{Swish}(x))$ | 01.00 (01.00 ± 0.00) | 01.00 (01.00 ± 0.00) | 75.06 (75.02 ± 0.35) ** |
| **General Activation Functions** | | | |
| $\max\{\text{Swish}(x), \alpha \log(\sigma(\text{ReLU}(x)))\}$ | 72.50 (72.54 ± 0.26) *** | 69.97 (69.91 ± 0.37) | 75.21 (75.20 ± 0.41) *** |
| $\min\{\text{Swish}(x), \alpha \text{ELU}(\text{ReLU}(\beta x))\}$ | 72.44 (72.39 ± 0.29) *** | 69.90 (69.82 ± 0.40) | 75.20 (75.27 ± 0.38) *** |
| $\log(\sigma(x))$ | 72.38 (72.33 ± 0.32) *** | 69.49 (69.58 ± 0.35) | 75.45 (75.53 ± 0.37) *** |
| **Baseline Activation Functions** | | | |
| ReLU | 71.44 (71.46 ± 0.50) | 69.78 (69.64 ± 0.65) | 74.43 (74.39 ± 0.44) |
| ELiSH | 01.00 (01.00 ± 0.00) | 01.00 (01.00 ± 0.00) | 75.16 (75.20 ± 0.31) *** |
| ELU | 72.41 (72.30 ± 0.32) *** | 69.59 (69.67 ± 0.46) | 74.86 (74.95 ± 0.30) ** |
| GELU | 72.00 (71.95 ± 0.35) * | 70.16 (70.19 ± 0.40) * | 74.84 (74.86 ± 0.33) ** |
| HardSigmoid | 55.55 (54.99 ± 1.00) | 33.31 (32.55 ± 4.06) | 65.03 (64.90 ± 0.69) |
| Leaky ReLU | 71.76 (71.73 ± 0.33) | 69.77 (69.78 ± 0.33) | 74.75 (74.73 ± 0.35) * |
| Mish | 72.02 (71.95 ± 0.41) * | 70.03 (69.88 ± 0.54) | 75.33 (75.32 ± 0.29) *** |
| SELU | 70.55 (70.53 ± 0.42) | 68.51 (68.52 ± 0.29) | 73.86 (73.79 ± 0.36) |
| sigmoid | 56.45 (56.10 ± 0.98) | 37.07 (36.47 ± 3.32) | 66.72 (66.45 ± 0.92) |
| Softplus | 72.25 (72.27 ± 0.26) *** | 69.71 (69.71 ± 0.36) | 75.47 (75.46 ± 0.52) *** |
| Softsign | 56.72 (56.30 ± 2.16) | 58.33 (58.38 ± 0.96) | 69.31 (69.33 ± 0.39) |
| Swish | 72.27 (72.26 ± 0.28) *** | 69.60 (69.68 ± 0.38) | 75.17 (75.08 ± 0.36) *** |
| tanh | 56.29 (56.52 ± 1.53) | 63.89 (63.88 ± 0.38) | 70.53 (70.44 ± 0.40) |
| **Parametric Baseline Functions** | | | |
| $\alpha \text{ReLU}(\beta x)$ | 72.01 (71.96 ± 0.31) ** | 68.91 (68.93 ± 0.22) | 73.60 (73.52 ± 0.37) |
| $\alpha \text{ELiSH}(\beta x)$ | 01.00 (01.00 ± 0.00) | 01.00 (01.00 ± 0.00) | 73.95 (73.94 ± 0.33) |
| $\alpha \text{ELU}(\beta x)$ | 71.96 (71.98 ± 0.24) ** | 68.91 (69.06 ± 0.37) | 74.03 (73.97 ± 0.45) |
| $\alpha \text{GELU}(\beta x)$ | 71.86 (71.96 ± 0.34) ** | 69.35 (69.39 ± 0.35) | 73.77 (73.83 ± 0.24) |
| $\alpha \text{HardSigmoid}(\beta x)$ | 66.74 (66.70 ± 0.64) | 33.47 (34.33 ± 6.53) | 65.09 (65.10 ± 0.40) |
| $\alpha \text{Leaky ReLU}(\beta x)$ | 71.70 (71.74 ± 0.39) | 69.18 (69.11 ± 0.47) | 73.53 (73.44 ± 0.29) |
| $\alpha \text{Mish}(\beta x)$ | 72.02 (72.11 ± 0.31) ** | 69.66 (69.51 ± 0.67) | 73.72 (73.72 ± 0.32) |
| $\alpha \text{SELU}(\beta x)$ | 71.04 (71.07 ± 0.33) | 68.06 (68.05 ± 0.39) | 73.44 (73.37 ± 0.38) |
| $\alpha \text{sigmoid}(\beta x)$ | 67.16 (66.98 ± 0.66) | 43.72 (44.40 ± 2.62) | 66.80 (66.98 ± 0.85) |
| $\alpha \text{Softplus}(\beta x)$ | 71.82 (71.73 ± 0.31) | 68.84 (68.84 ± 0.30) | 73.92 (73.95 ± 0.37) |
| $\alpha \text{Softsign}(\beta x)$ | 62.19 (62.12 ± 0.83) | 01.00 (9.18 ± 13.75) | 68.91 (68.87 ± 0.38) |
| $\alpha \text{Swish}(\beta x)$ | 72.36 (72.26 ± 0.29) *** | 69.25 (69.25 ± 0.28) | 73.97 (73.93 ± 0.22) |
| $\alpha \tanh(\beta x)$ | 63.72 (63.55 ± 0.56) | 01.00 (02.92 ± 6.07) | 69.61 (69.55 ± 0.62) |
| PReLU | 72.25 (72.23 ± 0.37) *** | 69.67 (69.77 ± 0.40) | 74.99 (75.10 ± 0.53) ** |
| $\text{PSwish} = x \cdot \sigma(\beta x)$ | 72.46 (72.40 ± 0.31) *** | 70.19 (70.16 ± 0.46) * | 75.37 (75.39 ± 0.28) *** |

using ReLU by default is suboptimal. The best baseline activation function is different for different architectures, reinforcing the importance of developing specialized activation functions.

Because PANGAEA uses validation accuracy from a single neural network to assign fitness scores to activation functions, there is selective pressure to discover functions that exploit the structure of the network. The functions thus become specialized to the architecture. They increase the performance of that architecture; however, they may not be as effective with other architectures. Specialized activation function accuracies are highlighted in gray in Table 2. To verify that the functions are customized to a specific architecture, the functions were cross-evaluated with other architectures.

PANGAEA discovered two specialized activation functions for WRN-10-4 and one for ResNet-v1-56 that achieved statistically significant improvements in mean accuracy over all baseline activation functions. All three specialized activation functions evolved for ResNet-v2-56 significantly outperformed ReLU as well. These results strongly demonstrate the power of customizing activation functions to architectures.

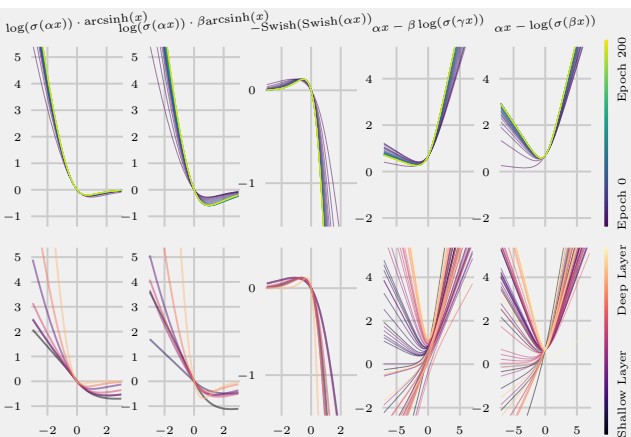

Figure 5: Adaptation of parametric activation functions over time and space. **Top:** The parameters change during training, resulting in different activation functions in the early and late stages. The plots were created by averaging the values of $\alpha$, $\beta$, and $\gamma$ across the entire network at different training epochs. **Bottom:** The parameters are updated separately in each channel, inducing different activation functions at different locations of a neural network. The plots were created by averaging $\alpha$, $\beta$, and $\gamma$ at each layer of the network after the completion of training.

**General Activation Functions** Although the best performance tends to come from specialization, it is also useful to discover activation functions that achieve high accuracy across multiple architectures. For instance, they could be used initially on a new architecture before spending compute on specialization. A powerful albeit computationally demanding approach would be to evolve general functions directly, by evaluating candidates on multiple architectures during evolution. However, it turns out that each specialized evolution run already generates a variety of functions, many of which are general.

To evaluate whether the PANGAEA runs discovered general functions as well, the top 10 functions from each run were combined into a pool of 30 candidate functions. Each candidate was assigned three fitness scores equal to the average validation accuracy from two independent training runs on each of the three architectures. Candidate functions that were Pareto-dominated, were functionally equivalent to one of the baseline activation functions, or had already been selected as a specialized activation function were discarded, leaving three Pareto-optimal general activation functions.

These functions indeed turned out to be effective as general activation functions: they all performed well on all architectures. One outperformed all baseline activation functions on WRN-10-4, while two functions on ResNet-v1-56 and three functions on ResNet-v2-56 outperformed 25 of the 28 baseline functions. However, specialized activation functions, i.e. those specifically evolved for each architecture, still tend to give the biggest improvements.

**Shapes of Discovered Functions** Many of the top discovered activation functions are compositions of multiple unary operators. These functions do not exist in the core unit search space of Ramachandran et al. (2018), which requires binary operators. They also do not exist in the $S_1$ or $S_2$ search spaces proposed by Bingham et al. (2020), which are too shallow. The design of the search space is therefore as important as the search algorithm itself. Previous search spaces that rely on repeated fixed building blocks only have limited representational power. In contrast, PANGAEA utilizes a flexible search space that can represent activation functions in an arbitrary computation graph.

Figure 5 shows examples of parametric activation functions discovered by PANGAEA. As training progresses, gradient descent makes small adjustments to the function parameters $\alpha$, $\beta$, and $\gamma$, resulting in activation functions that change over time. This result suggests that it is ad-

Table 3: CIFAR-100 test set accuracy shown as a median of ten runs, with mean $\pm$ sample standard deviation in parenthesis. The parametric evolved functions tend to outperform their non-parametric counterparts, demonstrating the value of parameterization.

| **WRN-10-4** | |
|---|---|
| $\log(\sigma(\alpha x)) \cdot \mathrm{arcsinh}(x)$ | **73.23** (73.16 ± 0.41) |
| $\log(\sigma(\alpha x)) \cdot \beta \mathrm{arcsinh}(x)$ | 73.22 (73.20 ± 0.37) |
| $\log(\sigma(x)) \cdot \mathrm{arcsinh}(x)$ | 72.42 (72.51 ± 0.30) |
| $-\mathrm{Swish}(\mathrm{Swish}(\alpha x))$ | **72.38** (72.49 ± 0.55) |
| $-\mathrm{Swish}(\mathrm{Swish}(x))$ | 71.99 (71.97 ± 0.22) |
| **ResNet-v1-56** | |
| $\alpha x - \beta \log(\sigma(\gamma x))$ | **70.82** (71.01 ± 0.64) |
| $\alpha x - \log(\sigma(\beta x))$ | 70.30 (70.30 ± 0.58) |
| $x - \log \sigma(x)$ | 69.44 (69.29 ± 0.45) |
| **ResNet-v2-56** | |
| $\min\{\log(\sigma(x)), \alpha \log(\sigma(\beta x))\}$ | 75.20 (75.19 ± 0.39) |
| $\log(\sigma(x))$ | **75.45** (75.53 ± 0.37) |

vantageous to have one activation function in the early stages of training when the network learns rapidly, and a different activation function in the later stages of training when the network is focused on fine-tuning. The parameters $\alpha$, $\beta$, and $\gamma$ are also learned separately for the different channels, resulting in activation functions that vary with location in a neural network. Functions in deep layers (near the output) are more nonlinear than those in shallow layers (closer to the input), possibly contrasting the need to form regularized embeddings with the need to form categorizations. In this manner, PANGAEA customizes the activation functions to both time and space for each architecture.

## 6 ABLATIONS AND VARIATIONS

**Effect of Parameterization** To understand the effect that parameterizing activation functions has on performance, the specialized functions (Table 2) were trained without them. As Table 3 shows, when parameters are removed, performance drops. The function $\log(\sigma(x))$ is the only exception to this rule, but its high performance is not surprising, since it was previously discovered as a general activation function (Table 2). These results confirm that the learnable parameters contributed to the success of PANGAEA.

**Search Strategy** As additional baseline comparisons, two alternative search strategies were used to discover activation functions for WRN-10-4. First, a random search baseline was established by applying random mutations without regard to fitness values. This approach corresponds to setting evolutionary parameters $P = 1$, $S = 1$, and $V = 0\%$. Second, to understand the effects of function parameterization, a nonparametric evolution baseline was run. This setting is identical to PANGAEA, except functions are not parameterized (Figure 3). Otherwise, both baselines follow the same setup as PANGAEA, including evaluating $C = 1,000$ candidate functions and reranking the most promising ones (Section 3.2).

Table 4 shows the results of this experiment. Random search is able to discover good functions that outperform ReLU, but the functions are not as powerful as those discovered by PANGAEA. This result demonstrates the importance of fitness selection in evolutionary search. The functions discovered by nonparametric evolution similarly outperform ReLU but underperform PANGAEA. Interestingly, without parameterization, evolution is not as creative: two of the three functions discovered are merely Swish multiplied by a constant. Random search and nonparametric evolution both discovered good functions that improved accuracy, but PANGAEA achieves the best performance by combining the advantages of fitness selection and function parameterization.

**Scaling Up** PANGAEA discovered specialized activation functions for WRN-10-4, ResNet-v1-56, and ResNet-v2-56. Table 5 shows the performance of these activation

Table 4: WRN-10-4 accuracy with different activation functions on CIFAR-100, shown as a median of ten runs, with mean $\pm$ sample standard deviation in parenthesis. PANGAEA discovers better activation functions than random search and nonparametric evolution.

| **PANGAEA** | |
|---|---|
| $\log(\sigma(\alpha x)) \cdot \operatorname{arcsinh}(x)$ | **73.23** (73.16 ± 0.41) |
| $\log(\sigma(\alpha x)) \cdot \beta\operatorname{arcsinh}(x)$ | 73.22 (73.20 ± 0.37) |
| $-\operatorname{Swish}(\operatorname{Swish}(\alpha x))$ | 72.38 (72.49 ± 0.55) |
| **Random Search** | |
| $\alpha\operatorname{Swish}(x)$ | 72.80 (72.85 ± 0.25) |
| $\operatorname{Softplus}(x) \cdot \arctan(\alpha x)$ | 72.78 (72.81 ± 0.35) |
| $\operatorname{ReLU}(\alpha\operatorname{arcsinh}(\beta\sigma(x))) \cdot \operatorname{SELU}(\gamma x)$ | 72.63 (72.69 ± 0.21) |
| **Nonparametric Evolution** | |
| $\cosh(1) \cdot \operatorname{Swish}(x)$ | 72.81 (72.78 ± 0.24) |
| $(e^1 - 1) \cdot \operatorname{Swish}(x)$ | 72.57 (72.52 ± 0.34) |
| $\operatorname{ReLU}(\operatorname{Swish}(x))$ | 72.06 (72.04 ± 0.54) |
| ReLU | 71.44 (71.46 ± 0.50) |
| Swish | 72.27 (72.26 ± 0.28) |

Table 5: Specialized activation functions discovered for WRN-10-4, ResNet-v1-56, and ResNet-v2-56 are evaluated on larger versions of those architectures: WRN-16-8, ResNet-v1-110, and ResNet-v2-110, respectively. CIFAR-100 test accuracy is reported as the median of three runs, with mean $\pm$ sample standard deviation in parenthesis. Specialized activation functions successfully transfer to WRN-16-8 and ResNet-v2-110, outperforming ReLU.

| **WRN-16-8** | |
|---|---|
| $\log(\sigma(\alpha x)) \cdot \operatorname{arcsinh}(x)$ | **78.42** (78.34 ± 0.20) |
| $\log(\sigma(\alpha x)) \cdot \beta\operatorname{arcsinh}(x)$ | 78.38 (78.36 ± 0.17) |
| $-\operatorname{Swish}(\operatorname{Swish}(\alpha x))$ | 77.90 (78.00 ± 0.35) |
| ReLU | 78.14 (78.15 ± 0.03) |
| **ResNet-v1-110** | |
| $\alpha x - \beta\log(\sigma(\gamma x))$ | 70.88 (70.85 ± 0.50) |
| $\alpha x - \log(\sigma(\beta x))$ | 70.40 (70.34 ± 0.60) |
| $\max\{\operatorname{Swish}(x), 0\}$ | 70.30 (70.36 ± 0.56) |
| ReLU | **71.15** (71.23 ± 0.25) |
| **ResNet-v2-110** | |
| $\operatorname{Softplus}(\operatorname{ELU}(x))$ | **77.34** (77.14 ± 0.38) |
| $\min\{\log(\sigma(x)), \alpha\log(\sigma(\beta x))\}$ | 76.99 (76.93 ± 0.19) |
| $\operatorname{SELU}(\operatorname{Swish}(x))$ | 77.04 (76.96 ± 0.14) |
| ReLU | 76.35 (76.34 ± 0.11) |

functions when paired with the larger WRN-16-8, ResNet-v1-110, and ResNet-v2-110 architectures. Due to time constraints, ReLU is the only baseline activation function in these experiments.

Two of the three functions discovered for WRN-10-4 outperform ReLU with WRN-16-8, and all three functions discovered for ResNet-v2-56 outperform ReLU with ResNet-v2-110. Interestingly, ReLU achieves the highest accuracy for ResNet-v1-110, where activation functions are part of the skip connections, but not for ResNet-v2-110, where they are not. Thus, it is easier to achieve

high performance with specialized activation functions on very deep architectures when they are not confounded by skip connections. Notably, ResNet-v2-110 with Softplus($ELU(x)$) performs comparably to much larger ResNet-v2-1001 with ReLU (77.34 vs. 77.29, as reported by He et al. (2016b)).

Evolving novel activation functions can be computationally expensive. The results in Table 5 suggest that it is possible to reduce this cost by evolving activation functions for smaller architectures, and then using the discovered functions with larger architectures.

**All-CNN-C** Finally, to verify that PANGAEA is effective with different datasets and types of architectures, activation functions were evolved for the All-CNN-C (Springenberg et al., 2015) architecture on the CIFAR-10 dataset. All-CNN-C is quite distinct from the architectures considered above: it contains only convolutional layers, activation functions, and a global average pooling layer, but it does not have residual connections. As shown in Table 6, PANGAEA improves significantly over ReLU in this setting as well. The accuracy improvement from 88.47% to 92.80% corresponds to an impressive 37.55% reduction in the error rate. This experiment provides further evidence that PANGAEA can improve performance for different architectures and tasks.

Table 6: All-CNN-C accuracy with different activation functions on CIFAR-10, shown as a median of ten runs, with mean $\pm$ sample standard deviation in parenthesis. PANGAEA improves performance significantly also with this different architecture and task.

| | |
|---|---|
| $\alpha\mathrm{ReLU}(\beta\|\mathrm{ReLU}(\gamma x)\|)$ | **92.80** (92.77 $\pm$ 0.13) |
| $\alpha\mathrm{Swish}(x) \cdot \cosh(\beta)$ | 92.67 (92.66 $\pm$ 0.08) |
| $\alpha\mathrm{Swish}(\beta x)$ | 92.63 (76.15 $\pm$ 34.86) |
| ReLU | 88.47 (88.47 $\pm$ 0.14) |

# 7 FUTURE WORK

It is difficult to select an appropriate activation function for a given architecture because the activation function, network topology, and training setup interact in complex ways. It is especially promising that PANGAEA discovered activation functions that significantly outperformed the baselines, since the architectures and training setups were standard and developed with ReLU. A compelling research direction is to jointly optimize the architecture, training setup, and activation function.

More specifically, there has been significant recent research in automatically discovering the architecture of neural networks through gradient-based, reinforcement learning, or neuroevolutionary methods (Elsken et al., 2019; Wistuba et al., 2019; Real et al., 2019). In related work, evolution was used discover novel loss functions automatically (Gonzalez & Miikkulainen, 2019; 2020; Liang et al., 2020), outperforming the standard cross entropy loss. In the future, it may be possible to optimize many of these aspects of neural network design jointly. Just as new activation functions improve the accuracy of existing network architectures, it is likely that different architectures will be discovered when the activation function is not ReLU. One such example is EfficientNet (Tan & Le, 2019), which achieved state-of-the-art accuracy for ImageNet (Deng et al., 2009) using the Swish activation function (Ramachandran et al., 2018; Elfwing et al., 2018). Coevolution of activation functions, topologies, loss functions, and possibly other aspects of neural network design could allow taking advantage of interactions between them, leading to further improvements in the future.

# 8 CONCLUSION

This paper introduced PANGAEA, a technique for automatically designing novel, high-performing, parametric activation functions. PANGAEA builds a synergy of two different optimization processes: evolutionary population-based search for the general form, and gradient descent-based fine-tuning of the parameters of the activation function. Compared to previous studies, the search space is extended to include deeper and more complex functional forms, including ones unlikely to be discovered by humans. The parameters are adapted during training and are different in different locations of the architecture, thus customizing the functions over both time and space. PANGAEA is able to discover general activation functions that perform well across architectures, and specialized functions taking advantage of a particular architecture, significantly outperforming previously proposed activation functions in both cases. It is thus a promising step towards automatic configuration of neural networks.

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

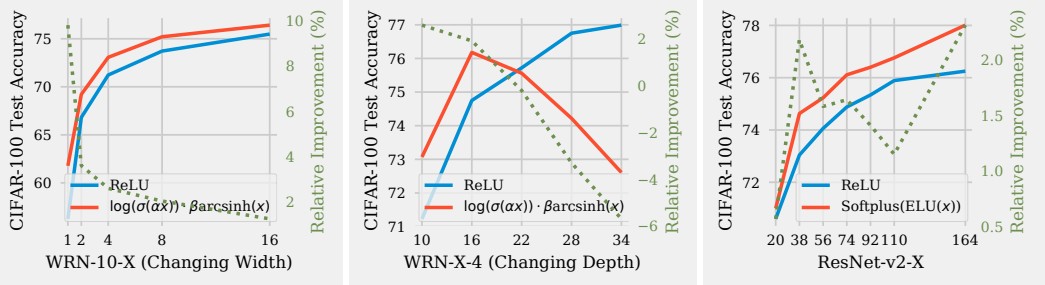

Figure 6: CIFAR-100 test accuracy for different neural networks and activation functions. Accuracy with ReLU is shown in blue, and accuracy with the specialized activation functions in red. The relative improvement of the specialized functions over ReLU is shown as a dotted green line, according to the axis values on the right of each plot. **Left:** The depth of Wide ResNet is fixed at 10, and the width varies from 1 to 16. **Center:** The depth of Wide ResNet varies from 10 to 34, while the width is fixed at four. **Right:** The depth of Preactivation ResNet ranges from 20 to 164. The width and depth of a network can affect how much a specialized activation function outperforms ReLU.

## A    ADJUSTING ARCHITECTURE WIDTH AND DEPTH

To further investigate the effect of network size on the performance of novel activation functions, two specialized activation functions were paired with neural networks of different widths and depths. Due to time constraints, the results in this experiment are based on single training runs.

**Wide Residual Networks**    The specialized activation function $\log(\sigma(\alpha x)) \cdot \beta \mathrm{arcsinh}(x)$ was discovered for a Wide ResNet of depth 10 and width four (WRN-10-4). Figure 6 shows the performance of this function when paired with Wide ResNets of different depths and widths.

For all widths tested, $\log(\sigma(\alpha x)) \cdot \beta \mathrm{arcsinh}(x)$ outperforms ReLU, albeit with diminishing returns as the width becomes large. This result implies that $\log(\sigma(\alpha x)) \cdot \beta \mathrm{arcsinh}(x)$ gives the network more representational power than ReLU. As the width of the architecture is increased, the additional network parameters partially offset this advantage, explaining the decreasing relative improvement of $\log(\sigma(\alpha x)) \cdot \beta \mathrm{arcsinh}(x)$ over ReLU.

For a fixed architecture width of four, $\log(\sigma(\alpha x)) \cdot \beta \mathrm{arcsinh}(x)$ outperforms ReLU only when the depth is 10 and 16. Surprisingly, as the depth is increased to 22 and beyond, the performance of $\log(\sigma(\alpha x)) \cdot \beta \mathrm{arcsinh}(x)$ drops. This result suggests that $\log(\sigma(\alpha x)) \cdot \beta \mathrm{arcsinh}(x)$ is specialized to shallow architectures.

**Preactivation Residual Networks**    The specialized activation function $\mathrm{Softplus}(\mathrm{ELU}(x))$ was discovered for a Preactivation ResNet of depth 56 (ResNet-v2-56). Figure 6 shows the performance of this function when paired with Preactivation ResNets of different depths. Unlike with the Wide ResNets, there is no clear increase or decrease in relative improvement over ReLU as depth increases. Impressively, ResNet-v2-164 with $\mathrm{Softplus}(\mathrm{ELU}(x))$ achieved test set accuracy 78.01, outperforming the accuracy of ResNet-v2-1001 with ReLU (77.29) as reported by He et al. (2016b).

## B    TRAINING DETAILS

**Wide Residual Network (WRN-10-4)**    When measuring final performance after evolution, the standard WRN setup is used; all ReLU activations in WRN-10-4 are replaced with the evolved activation function, but no other changes to the architecture are made. The network is optimized using stochastic gradient descent with Nesterov momentum 0.9. The network is trained for 200 epochs; the initial learning rate is 0.1, and it is decreased by a factor of 0.2 after epochs 60, 120, and 160. Dropout probability is set to 0.3, and L2 regularization of 0.0005 is applied to the weights. Data augmentation includes featurewise center, featurewise standard deviation normalization, horizontal flip, and random $32 \times 32$ crops of images padded with four pixels on all sides. This setup was chosen to mirror the original WRN setup (Zagoruyko & Komodakis, 2016) as closely as possible.

During evolution of activation functions, the training is compressed to save time. The network is trained for only 100 epochs; the learning rate begins at 0.1 and is decreased by a factor of 0.2 after epochs 30, 60, and 80. Empirically, the accuracy achieved by this shorter schedule is sufficient to guide evolution; the computational cost saved by halving the time required to evaluate an activation function can then be used to search for additional activation functions.

**Residual Network (ResNet-v1-56)**  As with WRN-10-4, when measuring final performance with ResNet-v1-56, the only change to the architecture is replacing the ReLU activations with an evolved activation function. The network is optimized with stochastic gradient descent and momentum 0.9. Dropout is not used, and L2 regularization of 0.0001 is applied to the weights. In the original ResNet experiments (He et al., 2016a), an initial learning rate of 0.01 was used for 400 iterations before increasing it to 0.1, and further decreasing it by a factor of 0.1 after 32K and 48K iterations. An iteration represents a single forward and backward pass over one training batch, while an epoch consists of training over the entire training dataset. In this paper, the learning rate schedule is implemented by beginning with a learning rate of 0.01 for one epoch, increasing it to 0.1, and then decreasing it by a factor of 0.1 after epochs 91 and 137. (For example, (48K iterations / 45K training images) * batch size of 128 $\approx$ 137.) The network is trained for 200 epochs in total. Data augmentation includes a random horizontal flip and random $32 \times 32$ crops of images padded with four pixels on all sides, as in the original setup (He et al., 2016a).

When evolving activation functions for ResNet-v1-56, the learning rate schedule is again compressed. The network is trained for 100 epochs; the initial warmup learning rate of 0.01 still lasts one epoch, the learning rate increases to 0.1, and then decreases by a factor of 0.1 after epochs 46 and 68. When evolving activation functions, their relative performance is more important than the absolute accuracies they achieve. The shorter training schedule is therefore a cost-efficient way of discovering high-performing activation functions.

**Preactivation Residual Network (ResNet-v2-56)**  The full training setup, data augmentation, and compressed learning rate schedule used during evolution for ResNet-v2-56 are all identical to those for ResNet-v1-56 with one exception: with ResNet-v2-56, it is not necessary to warm up training with an initial learning rate of 0.01 (He et al., 2016b), so this step is skipped.

**All-CNN-C**  When measuring final performance with All-CNN-C, the ReLU activation function is replaced with an evolved one, but the setup otherwise mirrors that of Springenberg et al. (2015) as closely as possible. The network is optimized with stochastic gradient descent and momentum 0.9. Dropout probability is 0.5, and L2 regularization of 0.001 is applied to the weights. The data augmentation involves featurewise centering and normalizing, random horizontal flips, and random $32 \times 32$ crops of images padded with five pixels on all sides. The initial learning rate is set to 0.01, and it is decreased by a factor of 0.1 after epochs 200, 250, and 300. The network is trained for 350 epochs in total.

During evolution of activation functions, the same training setup was used. It is not necessary to compress the learning rate schedule as was done with the residual networks because All-CNN-C trains more quickly.

**CIFAR-10**  As with CIFAR-100, a balanced validation set was created for CIFAR-10 by randomly selecting 500 images from each class, resulting in a training/validation/test split of 45K/5K/10K images.

## C  IMPLEMENTATION AND COMPUTE REQUIREMENTS

High-performance computing in two clusters is utilized for the experiments. One cluster uses HTCondor (Thain et al., 2005) for scheduling jobs, while the other uses the Slurm workload manager. Training is executed on GeForce GTX 1080 GPUs on both clusters. When a job begins executing, a parent activation function is selected by sampling $S = 16$ functions from the $P = 64$ most recently evaluated activation functions. This is a minor difference from the original regularized evolution (Real et al., 2019), which is based on a strict sliding window of size $P$. This approach may give extra influence to some activation functions, depending on how quickly or slowly jobs are executed in each

of the clusters. In practice the method is highly effective; it allows evolution to progress quickly by taking advantage of extra compute when demand on the clusters is low.

It is difficult to know ahead of time how computationally expensive the evolutionary search will be. Some activation functions immediately result in an undefined loss, causing training to end. In that case only a few seconds have been spent and another activation function can immediately be evaluated. Other activation functions train successfully, but their complicated expressions result in longer-than-usual training times. In these experiments, evolution for WRN-10-4 took 2,314 GPU hours, evolution for ResNet-v1-56 took 1,594 GPU hours, and evolution for ResNet-v2-56 took 2,175 GPU hours. These numbers do not include costs for reranking and repeated runs in the final experiments. Although substantial, the computational cost is negligible compared to the cost in human labor in designing activation functions. Evolution of parametric activation functions requires minimal manual setup and delivers automatic improvements in accuracy.

## D    BASELINE ACTIVATION FUNCTION DETAILS

Table 7: Baseline activation functions from the operator search space (Table 1) and final results (Table 2).

| Name | Definition | Reference(s) |
|---|---|---|
| ReLU | $\max\{x, 0\}$ | Nair & Hinton (2010) |
| ELiSH | $\frac{x}{1+e^{-x}}$ if $x \geq 0$ else $\frac{e^x-1}{1+e^{-x}}$ | Basirat & Roth (2018) |
| ELU | $x$ if $x \geq 0$ else $\alpha(e^x - 1)$, with $\alpha = 1$ | Clevert et al. (2015) |
| GELU | $x\Phi(x)$, with $\Phi(x) = P(X \leq x), X \sim \mathcal{N}(0,1)$, approximated as $0.5x(1 + \tanh[\sqrt{2/\pi}(x + 0.044715x^3)])$ | Hendrycks & Gimpel (2016) |
| HardSigmoid | $\max\{0, \min\{1, 0.2x + 0.5\}\}$ | |
| Leaky ReLU | $x$ if $x \geq 0$ else $0.01x$ | Maas et al. (2013) |
| Mish | $x \cdot \tanh(\text{Softplus}(x))$ | Misra (2019) |
| SELU | $\lambda x$ if $x \geq 0$ else $\lambda\alpha(e^x - 1)$, with $\lambda = 1.05070098, \alpha = 1.67326324$ | Klambauer et al. (2017) |
| sigmoid | $(1 + e^{-x})^{-1}$ | |
| Softplus | $\log(e^x + 1)$ | |
| Softsign | $x/(|x| + 1)$ | |
| Swish | $x \cdot \sigma(x)$, with $\sigma(x) = (1 + e^{-x})^{-1}$ | Ramachandran et al. (2018) and Elfwing et al. (2018) |
| tanh | $\frac{e^x - e^{-x}}{e^x + e^{-x}}$ | |
| PReLU | $x$ if $x \geq 0$ else $\alpha x$, where $\alpha$ is a per-neuron learnable parameter initialized to 0.25 | He et al. (2015) |
| PSwish | $x \cdot \sigma(\beta x)$, where $\beta$ is a per-channel learnable parameter | Ramachandran et al. (2018) |

