# OpenReview forum: "Discovering Parametric Activation Functions"
_ICLR.cc/2021/Conference — Reject_

### Official Review · AnonReviewer2 · 2020-10-20
**Solid paper with some unclarities in particular to contributions**

**Rating:** 6
**Confidence:** 5

**Review:**

The authors propose to search for activation functions with regularized evolution, an evolutionary algorithm proposed by Real et al. Various mutations are proposed that allow to investigate a larger search space than prior work. In particular, a mutation is added which adds trainable parameters to the activation function. The discovered activation functions are compared on three different architectures to several state-of-the-art activation functions.

The authors clearly describe their method. The idea and method are well-motivated and leave no scope for criticism. Experimental results indicate that the discovered activation functions are significantly better than a wide range of alternatives. Many related works are discussed, the authors might consider adding [1] to their discussion.
Currently unclear are the contributions of this work. In my understanding the authors claim to use different mutations and a different search space. Given the title, the main contribution seems to be the addition of parameters to the activation function. However, this idea is not new. It is e.g. used for PReLU but also the related work by Ramachandran et al. considers the option to add parameters as a unary operator. The authors attempt to study the influence of adding these parameters to the search space by investigating how the effective learning rate changes during training and at various depths. It would be interesting to see how the method performs without the proposed addition in section 3.2. Do any preliminary results indicate that this is significantly improving the results?
Very related to this: in the appendix you define the swish activation as x*\sigma(x). It would be interesting to know whether the authors used this function or the parametric version proposed by the original authors defined as x*\sigma(\beta * x).

The authors investigate whether the activation function generalize across architectures and notice that many do not. It would be interesting to know whether the discovered activation functions generalize to different data sets. The authors claimed that CIFAR-100 is more difficult than CIFAR-10. How do the transferred activation functions compare in this case to the baselines?

The generalization is only considered on ResNets. How about entirely different architectures such as some without residual connections, DenseNets, NASNet, EfficientNet or MobileNet? Will it also transfer to recurrent neural networks and other sorts of non-image architectures?

Concluding, the paper is well-written and seems to provide a solid empirical evaluation. I think the authors could be more explicit about the novelties presented in this work and why they matter.


[1] Jin-Young Kim, Sung-Bae Cho: Evolutionary Optimization of Hyperparameters in Deep Learning Models. CEC 2019: 831-837

**After Authors' Response:**
In my understanding the main contribution of this work is the identification of a new search space for activation functions. This might be very similar to earlier ones but the authors convinced me that it is a useful contribution. All my comments were addressed and as far as I saw also the comments of all other reviewers were addressed. For this reason, I increased my score.

---

> ### Author Response · Authors · 2020-11-24
> **Response to AnonReviewer2 comments**
>
>
>
> **Concern**: “the main contribution seems to be the addition of parameters to the activation function. However, this idea is not new.”
>
> **Response**: We don’t claim that parameterizing activation functions is a new idea.  As you mention, PReLU and parametric Swish are existing parametric activation functions, and we have added these to our list of baseline functions.  Our main contribution is a method for discovering novel parametric activation functions automatically with evolution---that is, new functions that take advantage of parameterization better than previous hand-designed and automatically discovered functions do.   Such a method is an advance in the state of the art.  For instance, a learnable $\beta$ parameter was a possible unary operator in the search space of Ramachandran et al., but their research did not explicitly focus on discovering parametric functions as ours does.  Similarly, the parametric version of Swish, $x \cdot \sigma(\beta x)$, did not consistently outperform the nonparametric version, $x \cdot \sigma(x)$.  Even in follow-up work by some of the same authors, the nonparametric version is used [2, 3]. In other words, although parametric functions have been used before, the functions have not been systematically optimized in this context. PANGAEA is a method that does that, and achieves better performance as a result.
>
>  ---
>
> **Question**: “It would be interesting to see how the method performs without the proposed addition in section 3.2.”
>
> **Response**: In the revision, this question is addressed in two ways.  First, the parameters of the top specialized activation functions discovered by PANGAEA were disabled; as a result, the performance dropped significantly.  Second, a full evolutionary search was run without parameterization.  This search discovered good functions that outperformed ReLU, but the functions were uncreative and did not outperform those discovered by PANGAEA.  (Refer to the two tables in the response to AnonReviewer4 above.)
>
> ---
>
> **Question**: “It would be interesting to know whether the authors used this function or the parametric version proposed by the original authors defined as x\sigma(\beta * x).”
>
> **Response**: When used as a baseline activation function and as a unary operator in the search space, Swish is defined as $x \cdot \sigma(x)$ (see the Appendix).  In addition, its parametric version, $x \cdot \sigma(\beta x)$, as well as a different version of it, $\alpha \textrm{Swish}(\beta x)$, were added to the baseline functions and evaluated as part of a larger set of parametric baselines. The results, shown in Table 2, demonstrate that such parameterization does not have a significant effect on their performance.
>
>   ---
>
> **Suggestion**: “It would be interesting to know whether the discovered activation functions generalize to different data sets...How about entirely different architectures such as some without residual connections?”
>
> **Response**: We applied PANGAEA to the All-CNN-C architecture on the CIFAR-10 dataset.  All-CNN-C is a very different architecture from the ones already included in the experiments: it contains only convolutional layers, activation functions, and a global average pooling layer, and no residual connections, as suggested. The results are consistent with those on CIFAR-100 and the other architectures.  PANGAEA improved accuracy from 88.47% with ReLU to 92.80% with an evolved function.  This corresponds to a 37.55% reduction in the error rate.
>
> ---
>
> **Suggestion**: “Will it also transfer to recurrent neural networks and other sorts of non-image architectures?”
>
> **Response**: This is an excellent suggestion which we will have to defer to future work.  We already see from this work that it is possible to specialize activation functions to different types of convolutional architectures.  It is probable that one can similarly specialize activation functions to recurrent architectures, and we are looking forward to testing this hypothesis.
>
> ---
>
>
> [2]  M. Tan and Q. Le.  Efficientnet: Rethinking model scaling for convolutional neural networks.  In International Conference on Machine Learning, pp. 6105–6114, 2019.
>
> [3]  C. Xie, M. Tan, B. Gong, A. Yuille, and Q. V. Le. Smooth adversarial training. arXiv:2006.14536, 2020.

---

> > ### Comment · AnonReviewer2 · 2020-11-24
> > **Thank you**
> >
> > I would like the authors for their clear reply. I have no further requests or questions. Thanks a lot for your effort.

---

### Official Review · AnonReviewer4 · 2020-10-25
**Missing the whole family of learnable activation functions**

**Rating:** 5
**Confidence:** 5

**Review:**

In this paper, the authors focus on the task of discovering activation functions via evolutionary methods. The paper is well written, easy to follow and technically sound.
My main concern with this paper is that the authors mention the importance of having activation functions learned for particular networks/tasks, yet failed to compare against any of the learnable activation functions that have the same objective.
Some of those activation functions are, e.g., APL [1], PAU [2], Mixtures of activations [3], and SReLU [4].

As a positive aspect, I see the benefit of your paper in that the given set of evolutionary operations might achieve some of the previous activation functions. However, your approach has an associated computational cost that is very high, and therefore it is very important to distill what makes it work and raise the following questions:

- Is it possible that the improvements come from the parametrization?.
- What is the behavior when you do exploration but not allowing parameters?.

One very interesting aspect of the functions that you learned, is that the form $log(\sigma(x))$ shows up repeatedly and in some cases including parameters. Mathematically, this is equivalent to $-SoftPlus(-x)$. The general form can be easily achieved by a parametric version $\alpha SoftPlus(\beta x)$, which appears often in table 2. Do you think this would be one big take-away message? The swish suggestion of your paper?
Since the SoftPlus is a soft version of the ReLU, one could also find interesting $\alpha ReLU (\beta x))$.

Indeed, your search algorithm is useful, but the results seem to point in the direction of parametric learnable activation functions, therefore a comparison to that family of activations is very important.


[1] F. Agostinelli, M. D. Hoffman, P. J. Sadowski, and P. Baldi. Learning activation functions to improve deep neural networks. In Workshop Track Proceedings of the International Conference on Learning Representations, 2015.

[2] A. Molina, P. Schramowski, K. Kersting. Pade Activation Units: End-to-end Learning of Flexible Activation Functions in Deep Networks. In International Conference on Learning Representations, 2020.

[3] F. Manessi and A. Rozza. Learning combinations of activation functions. In 2018 24th International Conference on Pattern Recognition (ICPR), pages 61–66. IEEE, 2018.

[4] X. Jin, C. Xu, J. Feng, Y. Wei, J. Xiong, and S. Yan. Deep learning with s-shaped rectified linear activation units. In Proceedings of the Thirtieth AAAI Conference on Artificial Intelligence, 2016.

[5] Ramachandran, P., Zoph, B. and Le, Q.V., 2017. Searching for activation functions. arXiv preprint arXiv:1710.05941.

---

> ### Author Response · Authors · 2020-11-24
> **Response to AnonReviewer4 comments**
>
>
>
> **Concern**: “My main concern with this paper is that the authors...failed to compare against any of the learnable activation functions...e.g., APL [1], PAU [2], Mixtures of activations [3], and SReLU [4].”
>
> **Response**: We appreciate you making us aware of these approaches.  To provide a comparison to baseline learnable activation functions, we have added PReLU and the parametric version of Swish, $x \cdot \sigma(\beta x)$, to Table 2.  The other functions are more complex, i.e. not simply plug-and-play but require significant implementation effort. We were not able to work on them during the rebuttal period but may be able to include them in the final version of the paper.
>
>  ---
>
> **Question**: “Is it possible that the improvements come from the parameterization?”
>
> **Response**: This is an interesting question.  To answer it, we trained all of the parametric specialized functions discovered by PANGAEA with the learnable parameters removed.  The results are included as Table 3 in the paper and below.  Notably, removing the parameters hurts performance, demonstrating the value of parameterization.  The only exception to this rule is $\log(\sigma(x))$, but this function was previously discovered as a general function by PANGAEA, so its high performance is not surprising.
>
>
> | WRN-10-4                                                 |                          | ResNet-v1-56                              |                          | ResNet-v2-56                                            |                          |
> |----------------------------------------------------------|--------------------------|-------------------------------------------|--------------------------|---------------------------------------------------------|--------------------------|
> | $\log(\sigma(\alpha x)) \cdot \textrm{arcsinh}(x)$       | $73.23 (73.16 \pm 0.41)$ | $\alpha x - \beta \log(\sigma(\gamma x))$ | $70.82 (71.01 \pm 0.64)$ | $\min\{\log(\sigma(x)), \alpha \log(\sigma(\beta x))\}$ | $75.20 (75.19 \pm 0.39)$ |
> | $\log(\sigma(\alpha x)) \cdot \beta \textrm{arcsinh}(x)$ | $73.22 (73.20 \pm 0.37)$ | $\alpha x - \log(\sigma(\beta x))$        | $70.30 (70.30 \pm 0.58)$ | $\log(\sigma(x))$                                       | $75.45 (75.53 \pm 0.37)$ |
> | $\log(\sigma(x)) \cdot \textrm{arcsinh}(x)$              | $72.42 (72.51 \pm 0.30)$ | $x - \log\sigma(x)$                       | $69.44 (69.29 \pm 0.45)$ |                                                         |                          |
> | $-\textrm{Swish}(\textrm{Swish}(\alpha x))$              | $72.38 (72.49 \pm 0.55)$ |                                           |                          |                                                         |                          |
> | $-\textrm{Swish}(\textrm{Swish}(x))$                     | $71.99 (71.97 \pm 0.22)$ |                                           |                          |                                                         |                          |
>
> ---
>
> **Question**: “What is the behavior when you do exploration but not allowing parameters?”
>
> **Response**: Good question; we were able to run this experiment for WRN-10-4 (for now, due to compute limitations).  The setup was identical to PANGAEA, including evaluating 1,000 candidate functions.  The results are included in Table 4 in the paper, and below.  Evolution without parameters discovers good functions that outperform ReLU, but the functions do not reach the performance of those discovered by PANGAEA.  Interestingly, removing the parameters makes evolution less creative: Two of the three functions are merely Swish multiplied by a constant.  This experiment therefore shows that parameterization is an important component of PANGAEA.
>
> | PANGAEA                                                  |                          | Nonparametric Evolution            |                          |
> |----------------------------------------------------------|--------------------------|------------------------------------|--------------------------|
> | $\log(\sigma(\alpha x)) \cdot \textrm{arcsinh}(x)$       | $73.23 (73.16 \pm 0.41)$ | $\cosh(1) \cdot \textrm{Swish}(x)$ | $72.81 (72.78 \pm 0.24)$ |
> | $\log(\sigma(\alpha x)) \cdot \beta \textrm{arcsinh}(x)$ | $73.22 (73.20 \pm 0.37)$ | $(e^1-1) \cdot \textrm{Swish}(x)$  | $72.57 (72.52 \pm 0.34)$ |
> | $-\textrm{Swish}(\textrm{Swish}(\alpha x))$              | $72.38 (72.49 \pm 0.55)$ | $\textrm{ReLU}(\textrm{Swish}(x))$ | $72.06 (72.04 \pm 0.54)$ |
>
>
>  ---
>
> **Suggestion**: “$\alpha \textrm{Softplus}(\beta x)$...one could also find interesting $\alpha \textrm{ReLU} (\beta x)$”
>
> **Response**: Interesting idea, and worth studying systematically. We created parametric versions of all baseline functions and added them to Table 2.  With the addition of PReLU and parametric Swish, this increases the number of baseline functions from 13 to 28.  In short, parameterization does not improve their performance significantly.

---

### Official Review · AnonReviewer3 · 2020-11-01
**With great intuitions come extra effort expectations**

**Rating:** 5
**Confidence:** 5

**Review:**

**Summary:** This paper proposes to use evolutionary computation to search for optimal activation functions tailored to a given architecture. The proposed PANGEA method practically re-implements a partial Genetic Programming setup applied to searching for neural network activation functions, with interesting (though minor) results. Further investigation is necessary.

**Quality:** The paper is well written, with a good structure. The literature review covers most basis, though with some crucial misses (GP and Random Guessing). The investigation presented is correct and through, though insufficient in breadth. The results are significant and interesting. The core of the paper is well put together, but it requires some more work.

**Clarity:** The concepts are explained clearly and well exposed. The results are motivated and discussed. On the actual work presented, I am happy with the clarity of exposition and presentation.

**Originality:** The work explores a highly promising direction, building on previous results but with a new twist. The results are bare-bone proof-of-concept, the baseline misses some foundation work, and the actual improvements are marginal at best.

**Pros:**
- Very interesting and promising research direction
- Correct experimentation methods
- Very well written and easy to read
- I personally find this approach highly promising and innovative, and fully support its investigation

**Cons:**
- While the direction taken is most intriguing, the actual quantity of study is relatively small
- The claims are not supported by sufficient experimentation (insufficient experiments)
- Some claims are not supported at all (tailoring activation to architecture)
- Missing referencing and comparison with GP.

**Comments:**
- Working with 1 dataset and 3 variations of 1 architecture is too small a baseline to prove the point. While I personally believe in the presented results, further experimentation is necessary.
- The paper exposure is very pleasant, but it effectively hides the relatively small size of the presented work. As a results it sounds redundant at time, with over-sized pictures and over-complete tables, while the Conclusion paragraph is shorter than the Future Work.
- Figures 4 and 5 are unnecessarily over-sized and need to be shrunk.
- The lack of mention of Genetic Programming is unacceptable, especially given the paper available space. Reinventing part of it and rebranding it as PANGEA  for the specific application is interesting but insufficient.
- To establish the performance of a highly randomized algorithm such as PANGEA, a pure-random parameter guessing baseline needs to be established. Under the proposed rebranding, that would be a run where the operator Regenerate has a 1.0 chance while the others have 0.0.
- The claim that the evolved activation function is tailored to the architecture is central to the work, but it is not proven. It should be shown with cross-evaluation of activation functions and architectures or removed from the paper.
- Might be an oversight on my part, but it is not clear how many activation functions were evolved for each architecture (3 are shown in the results table) and how their performance is aggregated (in the results table, it is not). As mainly randomized algorithms, evolutionary methods can converge to very different results over subsequent runs, especially in the presence of highly multimodal, indirectly computed fitness landscapes such as this. The method needs then to be evaluated as a whole, not the single produced results, as the performance eventually depends on the number of reruns (tending to infinity).
- Paragraph 3.2: while Genetic Algorithms and related Genetic Programming are indeed discrete search algorithms, state-of-the-art Evolution Strategies are excellent continuous optimization algorithms. Please fix the claim accordingly.
- First paragraph in 3.3: while reading up on ES, please adopt the established $(\mu,\lambda)$ and $(\mu+\lambda)$ terminology in place of the introduced $P$ and $S$.
- The code is not published as open source, limiting distribution and reproducibility.

**Final remarks:**
The number of papers crossing and integrating SGD (or DL) and EC (or NE) is growing over the years. This is very good news for the field, but makes it important that papers exploring new foundation directions such as this one are incontestable and thoroughly proven. If the paper is accepted to ICLR21, the authors have my congratulations and best wishes; but if not, rather than just resubmitting the same content to another conference, I urge them to expand their work beyond reproach, as this direction (and the proven quality of your experimentation and writing) has the potential for an absolutely top-quality publication.

---

> ### Author Response · Authors · 2020-11-24
> **Response 1 to AnonReviewer3 comments**
>
> **Concern**: "Working with 1 dataset and 3 variations of 1 architecture is too small a baseline"
>
> **Response**: We added experiments transferring the activation functions to scaled up architectures: from WRN-10-4, ResNet-v1-56, and ResNet-v2-56 to WRN-16-8, ResNet-v1-110, and ResNet-v2-110, respectively.  We also applied PANGAEA to the All-CNN-C architecture on CIFAR-10.  In total, our experiments now include 2 datasets, 7 architectures, and 52 different activation functions.
>
> ---
>
> **Concern**: “Figures 4 and 5 are unnecessarily over-sized”
>
> **Response**: The figures have been shrunk.
>
>  ---
>
> **Concern**: “The lack of mention of Genetic Programming is unacceptable”
>
> **Response**: Fixed.
>
>  ---
>
> **Concern**: “a pure-random parameter guessing baseline needs to be established...where the operator Regenerate has a 1.0 chance while the others have 0.0”
>
> **Response**: This is an excellent idea.  However, if regenerate is the only mutation, then the computation graph structure cannot change; only the unary and binary operators can.  This limitation means that the search space becomes restricted. Either it consists of computation graphs like the initial population ones in Figure 1, which is not very interesting, or something hand-designed to make it more powerful, which would make an unfair comparison to PANGAEA which does not assume such knowledge.
>
> Instead, to provide a fair random search baseline, we applied random mutations without regard to fitness value.  This approach corresponds to setting evolutionary parameters $P=1$, $S=1$, and $V=0\%$; otherwise the setup is identical to PANGAEA.  Because of the computational cost, we evaluated the random baseline on WRN-10-4 only for now.  The results are included in Table 4 in the paper, as well as below.  The conclusion is that random search does discover good activation functions that outperform ReLU, but it is not as powerful as PANGAEA.
>
> | PANGAEA                                                  |                          | Random Search                                                                           |                          |
> |----------------------------------------------------------|--------------------------|-----------------------------------------------------------------------------------------|--------------------------|
> | $\log(\sigma(\alpha x)) \cdot \textrm{arcsinh}(x)$       | $73.23 (73.16 \pm 0.41)$ | $\alpha \textrm{Swish}(x)$                                                              | $72.80 (72.85 \pm 0.25)$ |
> | $\log(\sigma(\alpha x)) \cdot \beta \textrm{arcsinh}(x)$ | $73.22 (73.20 \pm 0.37)$ | $\textrm{Softplus}(x) \cdot \arctan(\alpha x)$ & $72.78$                                | $72.78 (72.81 \pm 0.35)$ |
> | $-\textrm{Swish}(\textrm{Swish}(\alpha x))$              | $72.38 (72.49 \pm 0.55)$ | $\textrm{ReLU}(\alpha \textrm{arcsinh}(\beta \sigma(x))) \cdot \textrm{SELU}(\gamma x)$ | $72.63 (72.69 \pm 0.21)$ |
>
>
> You specifically suggested setting the probability of a regenerate mutation to 1.0, and we ran this experiment as well.  Utilizing only the initial computation graph structures from Figure 1, unary1(unary2(x)) and binary(unary1(x), unary2(x)), would be too restrictive.  Instead, we used computation graphs of the form binary(unary1(unary2(unary3(x))), unary4(x)), since this is the simplest fixed computation graph structure that includes all of the functions discovered by PANGAEA within its search space.  As mentioned above, this setting is already biased since the search space was designed with knowledge discovered with evolution.  Otherwise the setup was identical to PANGAEA.  Again, because of the computational cost, this experiment was run on WRN-10-4 only for now, with the results shown in the table below:
>
> | Random Search in the Focused Search Space (WRN-10-4)                                           |                                          |
> |------------------------------------------------------------------------------------------------|------------------------------------------|
> | $\alpha \textrm{HardSigmoid} (\beta \textrm{ELU}(\arctan (x))) \cdot \textrm{Swish}(\gamma x)$ | $73.35 (73.30 \pm 0.46)$ |
> | $\alpha \textrm{Softsign}(\textrm{Swish}(\textrm{Swish}(x))) - \textrm{Swish}(x)$              | $73.01 (73.03 \pm 0.19)$ |
> | bessel_i1e$(\|x\|) - \textrm{ReLU}(x)$                                               | $72.38 (72.35 \pm 0.21)$ |
>
>
> In this more focused search space even random search is able to discover powerful functions.  They are not statistically better than those discovered by PANGAEA, but they are better than ReLU. The conclusion is that much like how activation functions can be specialized to architectures, search spaces can be too.   To further explore this idea, we are currently evolving activation functions in this search space.  Once the evolution run is complete we plan to include the results in the final camera-ready version.

---

> ### Author Response · Authors · 2020-11-24
> **Response 2 to AnonReviewer3 comments**
>
>
>
> **Concern**: “The claim that the evolved activation function is tailored to the architecture is central to the work, but it is not proven. It should be shown with cross-evaluation of activation functions and architectures or removed from the paper.”
>
> **Response**: The results in Table 2 address this claim specifically.  The functions specialized to WRN-10-4 perform very well with WRN-10-4.  They are also cross-evaluated with ResNet-v1-56 and ResNet-v2-56, where they perform poorly.  The functions specialized to ResNet-v1-56 and ResNet-v2-56 are similarly cross-evaluated with the other architectures, with similar results.  Some table cells are also shaded in gray to illustrate that the function is specialized to the architecture. The text has been revised to make this point clear.
>
>  ---
>
> **Concern**: “it is not clear how many activation functions were evolved for each architecture”
>
> **Response**: The approach is described in Section 3.2: for each architecture $C=1{,}000$ functions are evaluated during the search, the top 10 of these are reranked, and the top three after reranking proceed to the final testing experiments.  These are the functions listed in the tables.
>
> ---
>
> **Concern**: “The method needs then to be evaluated as a whole, not the single produced results, as the performance eventually depends on the number of reruns”
>
> **Response**: You are correct.  In practice, since meta-learning experiments are computationally expensive, it is necessary to use the computational budget in a most informative way.  We could have chosen to run PANGAEA several times for one architecture and dataset to get a very good understanding of its performance distribution in a specific domain.  Instead, we chose to evaluate multiple architectures and datasets once.  Although independent runs will surely produce some variation, PANGAEA consistently outperforms the baseline functions on many different tasks, suggesting that the results are reliable.
>
>  ---
>
> **Concern**: “Paragraph 3.2: while Genetic Algorithms and related Genetic Programming are indeed discrete search algorithms, state-of-the-art Evolution Strategies are excellent continuous optimization algorithms. Please fix the claim accordingly.”
>
> **Response**: Note that this paragraph does not claim that continuous evolutionary optimization is not possible, it just lists gradient descent as a synergetic approach.  Since we have two optimization processes at play, the goal of the paragraph is to characterize the contributions of each: evolution discovers the computation graph structure, and gradient descent updates the learnable parameters during backpropagation. Other methods could in principle be used in these roles as well.
>
>  ---
>
> **Concern**: “please adopt the established $(\mu, \lambda)$ and $(\mu + \lambda)$ terminology in place of the introduced $P$ and $S$
>
> **Response**: That notation is possible if you insist; however, we chose to use $P$ and $S$ for consistency with Real et al. [1] who introduced regularized evolution.
>
>  ---
>
> **Concern**: “The code is not published as open source, limiting distribution and reproducibility.”
>
> **Response**: We are working on sorting out intellectual property and licensing issues.  We have created an anonymous Google Drive account with an empty folder accessible with this link: [https://drive.google.com/drive/folders/1XnKXRGeYqL_DSh01d8fOldxsFEO5x_RU?usp=sharing](https://drive.google.com/drive/folders/1XnKXRGeYqL_DSh01d8fOldxsFEO5x_RU?usp=sharing).  Once the code is ready we will upload it there.
>
> ---
>
> [1]  E. Real, A. Aggarwal, Y. Huang, and Q. V. Le. Regularized evolution for image classifier architecture search. In Proceedings of the aaai conference on artificial intelligence, volume 33, pp. 4780–4789, 2019.

---

### Decision · Program_Chairs · 2021-01-07
**Final Decision**

**Decision:**

Reject

**Comment:**

The paper proposes the idea of searching parameterized activation functions, in contrast to the previous handcraft or learnable ones. It may be a counterpart of neural architecture search.

Pros:
1. The idea is very interesting.
2. The paper is well written.
3. The experiments show improvements over baseline activation functions.

Cons:
1. The AC fully agreed with Reviewer #4 that the whole literature of learnable activation function is neglected (Reviewer #2 also alluded to this issue). Although the authors added experiments with learnable baseline activation functionss, the literature review on learnable activation function was not included accordingly.
2. Although the idea of searching activation functions is interesting, the AC doubted the necessity. Since the rich literature of learnable activation functions is already there (note that it is more than introducing parameters to handcrafted ones), can we simply learn piecewise linear activation functions with more pieces so that it can approximate complex enough functions? This can be much more easily implemented (can go along with weight training on the standard deep learning platform) and the computation cost will be much lower. Such a comparison is absolutely necessary.
3. The AC was actually worried about the activation functions founded as they may be too complex, so the generalization issue (even numerical stability issue) may be a concern. More thorough testing is necessary (currently only tested on CIFAR-100 and three CNNs; and Reviewers #3 and #2 also concerned about this issue).

Although Reviewer #2 raised his/her score, the final average score is still below threshold. So the AC decided to reject the paper.